# Structural and Functional Effects of the Interaction Between an Antimicrobial Peptide and Its Analogs with Model Bacterial and Erythrocyte Membranes

**DOI:** 10.3390/biom15081143

**Published:** 2025-08-07

**Authors:** Michele Lika Furuya, Gustavo Penteado Carretero, Marcelo Porto Bemquerer, Sumika Kiyota, Magali Aparecida Rodrigues, Tarcillo José de Nardi Gaziri, Norma Lucia Buritica Zuluaga, Danilo Kiyoshi Matsubara, Marcio Nardelli Wandermuren, Karin A. Riske, Hernan Chaimovich, Shirley Schreier, Iolanda Midea Cuccovia

**Affiliations:** 1Departamento de Bioquímica, Instituto de Química, Universidade de São Paulo, São Paulo 05508-220, Brazil; michelelika@usp.br (M.L.F.); gpbc@ruc.dk (G.P.C.); magarodrig@gmail.com (M.A.R.); tarcillo98gaziri@gmail.com (T.J.d.N.G.); luciabz@iq.usp.br (N.L.B.Z.); danilokiyoshi@usp.br (D.K.M.); hchaimo@usp.br (H.C.); 2Department of Science and Environment, Roskilde University, 4000 Roskilde, Denmark; 3Embrapa Gado de Leite, Juiz de Fora, Minas Gerais 36038-330, Brazil; marcelo.bemquerer@embrapa.br; 4Laboratório de Bioquímica de Proteínas e Peptídeos, Instituto Biológico, São Paulo 04014-002, Brazil; skiyota07@gmail.com; 5Instituto de Química, Universidade de São Paulo, São Paulo 05508-220, Brazil; nardelli@iq.usp.br; 6Departamento de Biofísica, Universidade Federal de São Paulo, São Paulo 04039-032, Brazil; kariske@unifesp.br

**Keywords:** antimicrobial peptides, model bacterial membranes, model red cell membranes, large unilamellar vesicles, giant unilamellar vesicles

## Abstract

Antimicrobial peptides (AMPs) are a primary defense against pathogens. Here, we examined the interaction of two BP100 analogs, R^2^R^5^-BP100 (where Arg substitutes Lys 2 and 5) and R^2^R^5^-BP100-A-NH-C_16_ (where an Ala and a C_16_ hydrocarbon chain are added to the R^2^R^5^-BP100 C-terminus), with membrane models. Large unilamellar vesicles (LUVs) and giant unilamellar vesicles (GUVs) were prepared with the major lipids in Gram-positive (GP) and Gram-negative (GN) bacteria, as well as red blood cells (RBCs). Fluorescence data, dynamic light scattering (DLS), and zeta potential measurements revealed that upon achieving electroneutrality through peptide binding, vesicle aggregation occurred. Circular dichroism (CD) spectra corroborated these observations, and upon vesicle binding, the peptides acquired α-helical conformation. The peptide concentration, producing a 50% release of carboxyfluorescein (C_50_) from LUVs, was similar for GP-LUVs. With GN and RBC-LUVs, C_50_ decreased in the following order: BP100 > R^2^R^5^-BP100 > R^2^R^5^BP100-A-NH-C_16_. Optical microscopy of GP-, GN-, and RBC-GUVs revealed the rupture or bursting of the two former membranes, consistent with a carpet mechanism of action. Using GUVs, we confirmed RBC aggregation by BP100 and R^2^R^5^-BP100. We determined the minimal inhibitory concentrations (MICs) of peptides for a GN bacterium (*Escherichia coli* (*E. coli*)) and two GP bacteria (two strains of *Staphylococcus aureus* (*S. aureus*) and one strain of Bacillus subtilis (*B. subtilis*)). The MICs for *S. aureus* were strain-dependent. These results demonstrate that Lys/Arg replacement can improve the parent peptide’s antimicrobial activity while increasing hydrophobicity renders the peptide less effective and more hemolytic.

## 1. Introduction

The discovery and widespread use of antibiotics have led to a significant increase in the average life expectancy of the world’s population, and they continue to be responsible for reducing mortality and morbidity from infectious diseases [1]. Inadequate use of antibiotics, widespread veterinary use, incomplete treatment of infections, and excessive consumption have contributed to the emergence of antibiotic-resistant bacteria [2].

Antimicrobial peptides (AMPs) are ubiquitous and constitute the primary line of defense against microbial invasion. Plants [3,4], vertebrates [5,6], invertebrates [5,6], and bacteria [7] are rich in AMPs. Nature-inspired AMPs may serve as alternatives to conventional antibiotics due to their ability to inhibit bacterial growth and destroy both Gram-positive (GP) and Gram-negative (GN) bacteria [8,9,10]. AMPs show a broad spectrum of action, and in addition to their effect on bacteria, they may be used against fungi, protozoa, and viral envelopes and are novel cytotoxic agents for cancer treatment [11].

Antibiotics typically target bacterial proteins, making it easier for genetic changes to lead to resistance. AMPs disrupt the cell membrane’s lipid bilayer structure, making it difficult for its components to be modified [12]. As the bactericidal effects of AMPs are primarily exerted on membranes, the selection of resistant mutants is rare [13].

Most AMPs are positively charged and amphipathic, with a net charge ranging from +2 to +6 at pH 7. Some AMPs are amphipathic α-helices with hydrophobic and hydrophilic faces [14]. Because of their net positive charge, AMPs can interact with and disturb bacterial cell membranes, which are rich in negatively charged lipids [15,16]. The hydrophilic face of alpha-helical AMPs, which contains polar and positively charged amino acid residues like arginine and lysine, and the hydrophobic face, which contains amino acid residues with non-polar side chains, are on opposite sides of the helix in the secondary structure of AMPs and other membrane-active peptides. The amphipathic alpha-helical nature of many AMPs is essential for their interaction with phospholipid membranes [16]. The action mechanisms of AMPs include (a) toroidal pore, (b) “barrel-stave” (barrel of stakes), and (c) carpet [17,18,19].

The peptide H-KKLFKKILKYL-NH_2_ (BP100), synthesized by Bardaji and coworkers [20], showed high antimicrobial activity, being less effective only when compared to streptomycin [21]. BP100 disrupts several membranes, showing greater selectivity to bacterial anionic membrane models [22].

In silico and CD structural studies of BP100 in phosphatidylcholine (PC) membrane models show that the peptide adopts a helical conformation [23,24,25]. Results from our group show that, after initial electrostatic binding, BP100 settles into the interface by inserting the hydrophobic face into the membrane [26]. The mechanism of action of BP100 on model membranes is dependent on the peptide-to-lipid ratio and the negative charge density of the membrane surface [24].

Here, we modified the BP100 sequence by replacing Lys^2^ and Lys^5^ with Arg (R^2^R^5^-BP100) to examine the effect of these substitutions on peptide conformation, membrane interaction, and antimicrobial activity. Both BP100 and R^2^R^5^-BP100 carboxyl-terminal groups are amidated. We chose positions 2 and 5, occupied by Lys in BP100, to maintain the same charge as that of BP100 BP100 and investigate the effectiveness of this mutation on peptide binding, as Arg’s guanidinium group binds more strongly to the phosphate group of phospholipids [27]. Indeed, the argument that rationalizes the Lys/Arg replacement has been used by G. Riesco-Llach and coworkers in a recent review on BP100 [28]. We also synthesized an analog of R^2^R^5^-BP100, where Ala was added at the C-terminus, followed by a C_16_ acyl chain (R^2^R^5^-BP100-A-NH-C_16_), thereby increasing the molecule’s overall hydrophobicity. Derivatives of peptides with long-chain fatty acids may lead not only to increments in their partition coefficients into bilayers, but also to changes in pharmacokinetic properties due to binding to serum albumin and transport to peripheral tissues. As demonstrated by Chen et al., changing the length of the hydrophobic chain on an AMP may change the antimicrobial effectiveness and hemolytic action [29]. For topical applications of AMPs, modification with a fatty acid or a long alkyl chain may contribute to the formation of subcutaneous deposits, thereby increasing the peptide’s half-life [30]. The peptide’s sequences and structures are shown in Figure 1.

Model membranes can be prepared with different lipid compositions, allowing for the mimicry of various bacterial and eukaryotic membranes [16]. The inner membrane of many bacteria contains phosphatidylethanolamine (PE) as the most prevalent zwitterionic phospholipid, along with negatively charged lipids such as phosphatidylglycerol (PG) and cardiolipin (CL). In general, PE is more abundant in membranes of GN, and PG is more abundant in membranes of GP bacteria. Still, there are some exceptions, such as *Clostridia* and *Bacillae*, which are GP and contain a high portion of PE [31,32,33].

Sialylated glycoproteins on the red blood cell (RBC) membrane are responsible for a negatively charged surface, which creates a repulsive electric zeta potential between cells [34]. Other mammalian membranes contain mainly phosphatidylcholine (PC), PE, sphingomyelin (SPH), phosphatidylserine (PS), and cholesterol (CHOL), distributed asymmetrically between the two monolayers of the membrane; the outer layer is rich in PC and sphingomyelin, corresponding to 45% and 42%, respectively, of the lipid composition [33,35].

We studied the interaction of BP100 and its analogs, R^2^R^5^-BP100 and R^2^R^5^-BP100-A-NH-C_16_, with different model membranes and their antimicrobial properties to understand the mechanisms that modulate the peptide’s interaction with target membranes.

We prepared large unilamellar vesicles (LUVs) and giant unilamellar vesicles (GUVs) mimicking the lipid composition of GP, GN bacteria, and human red blood cells [36]. The peptides’ secondary structure in aqueous solution and in the presence of GP, GN, and RBC-LUVs was determined by CD spectroscopy, and peptide binding to vesicles was assessed by fluorescence spectroscopy. Peptide-induced LUV permeabilization was monitored by measuring the leakage of a fluorescent probe. The effect of peptide binding on vesicle size, aggregation, and surface potential was evaluated by dynamic light scattering (DLS). We analyzed the effect of the peptides on the stability of GUVs using optical microscopy. The peptide’s biological activity was assessed by measuring the minimum inhibitory concentration (MIC) against GP and GN bacterial species as well as RBC hemolysis.

Our results clarify the mechanism of membrane destabilization by Lys/Arg replacement on BP100 and show that an increase in hydrophobicity enhances the efficiency of AMPs to disrupt lipid membrane models and, at the same time, prove that an increase in hydrophobicity in BP100 does not increase the antimicrobial activity of this peptide and enhances the extent of hemolysis of RBC.

## 2. Materials and Methods

### 2.1. Materials

The lipids (1-palmitoyl-2-oleoyl-*sn*-glycero-3-phosphocholine (POPC); 1-palmitoyl-2-oleoyl-*sn*-glycero-3-phospho-(1′-rac-glycerol) (POPG); 1-palmitoyl-2-oleoyl-*sn*-glycero-3-phosphoethanolamine (POPE); 1′,3′-bis [1-palmitoyl-2-oleoyl-*sn*-glycero-3-phospho]-glycerol (cardiolipin, CL); N-palmitoyl-D-erythrosphingosylphosphorylcholine (sphingomyelin, SPH); and cholesterol (CHOL)) were obtained from Avanti Polar Lipids. 5-(6)-Carboxyfluorescein (CF), obtained from Sigma-Aldrich, St. Louis, MO, USA, was purified using the method described by Manzini et al. [24]. Mueller–Hinton broth was obtained from KASVI. Amino acid derivatives and other reagents used for peptide synthesis were obtained from either Peptides International (Louisville, KY, USA) or Sigma-Aldrich. N,N-dimethylformamide was distilled before use.

### 2.2. Methods

#### 2.2.1. Peptide Synthesis

BP100 (KKLFKKILKYL-NH_2_), R^2^R^5^-BP100 (KRLFRKILKYL-NH_2_), and R^2^R^5^-BP100-A-NH-C_16_ (KRLFRKILKYLA-NH-C_16_H_33_) (Figure 1) were synthesized and purified as described previously [37]. Molecular weights and charges at pH 7.0 are shown in Table 1.

#### 2.2.2. Peptide Solutions

Peptide solutions of 1.0 mM were prepared by weighing the dry peptide and then solubilizing it in Milli-Q water. The final peptide concentration was measured using a Nanodrop N-100 spectrophotometer (Thermo Fisher Scientific, Wilmington, DE, USA), using the tyrosine absorption wavelength at 275 nm, the molar absorptivity coefficient at 275 nm (ε^275^ = 1400 M^−1^ × cm^−1^), and an optical path of 1.00 mm.

#### 2.2.3. Model Membrane Composition and Preparation of Large Unilamellar Vesicles (LUVs)

The lipid composition of model membranes used to mimic bacterial membranes was based on the composition of cytoplasmic membranes of GP and GN bacteria [31] of three types of lipids: POPE, CL, and POPG. RBC-LUV membranes were prepared with four lipids with no net charge: CHOL, POPC, SPH, and POPE [33,35] (Table 2).

From here onwards, the abbreviations GP, GN, and RBC will be used to refer to the model membranes corresponding to Gram-positive bacteria, Gram-negative bacteria, and red blood cells, respectively.

Phospholipid stock solutions were prepared in chloroform and quantified by determining the phosphate concentration [38]. Lipid films were prepared by mixing aliquots of the stock solutions in a glass tube, followed by solvent evaporation under a stream of nitrogen. The films were then dried under vacuum for at least one hour. Large unilamellar vesicles, LUVs, were obtained by resuspending the dry lipid film in 10 mM Tris-HCl, pH 7.4, buffer, to a total lipid concentration of 15 mM, followed by extrusion through polycarbonate membranes with a pore size of 100 nm (Nuclepore, Maidstone, UK).

#### 2.2.4. Fluorescence Studies of Peptide–Membrane Binding

Peptide fluorescence spectra were obtained using a Hitachi F-7000 spectrofluorometer (Hitachi, Tokyo, Japan), exciting the peptides’ Tyr residue at 275 nm and recording the emission between 280 nm and 400 nm. The peptide concentration was 20 μM, and the lipid concentration varied from 0 to 0.80 mM. To achieve the desired lipid concentrations in 450 μL peptide-containing samples, aliquots of 15 mM extruded vesicle stock solutions were added. For the calculation of the lipid/peptide ratios, the concentrations of peptide and lipid were corrected for dilution.

Peptide affinity for the lipid vesicles was calculated assuming a two-state model (free and bound peptides). Variation in the maximum fluorescence emission was normalized between 0 (peptide in solution) and 1 (fully bound) to reflect the fraction of bound peptide. The bound peptide fraction was calculated using Equation (1):

Fraction bound = (I_B_ − I_0_)/(I_MAX_ − I_0_)
(1)
 where I_0_, I_B_, and I_MAX_ are the fluorescence intensities at 305 nm, in buffer (I_0_), at a given lipid concentration (I_B_), and at the binding saturation concentration (I_MAX_). The normalized fluorescence for each LUV addition was estimated as a function of the [Lipid]/[Peptide], i.e., [L]/[P] ratio.

#### 2.2.5. Circular Dichroism (CD) of Peptide–Membrane Interaction

The peptides’ secondary structures were investigated using circular dichroism (CD) in solution and in the presence of model membranes with varying lipid compositions. CD spectra were obtained with a Jasco J-720 spectropolarimeter (Jasco, Tokyo, Japan) at room temperature using quartz cells of 1.00 mm optical length (or 0.1 mm, see Appendix A). Spectra were collected in the far-UV range between 190 and 260 nm; samples consisted of 20 μM peptide in 10 mM Tris-HCl buffer, pH 7.4, and in the presence of fixed LUV concentrations (0.125 mM GP, 0.500 mM GN, and 1.00 mM RBC). Aliquots of the extruded vesicle stock solutions (15 mM total lipid) were added to 300 μL of peptide solutions. CD spectra of the buffer and lipid suspensions were obtained under the same conditions as those of the peptide-containing samples. The final spectrum is the average of six accumulations after subtraction of the spectra of samples without peptide and correction for volume variation, as LUV aliquots were added to the sample.

#### 2.2.6. Evaluation of Vesicle Size, Polydispersity, and Zeta Potential: Dynamic Light Scattering (DLS)

GP, GN, and RBC-LUVs were prepared in 10 mM Tris-HCl buffer, pH 7.4, and the vesicle properties were measured using dynamic light scattering equipment, a Malvern Zetasizer Nano apparatus equipped with a 633 nm laser (Malvern Worcestershire, UK). The Malvern software assigned a value of 1.0 to the Henry function. The LUVs’ zeta potential was calculated from the electrophoretic mobility using Henry’s equations:
(2)$$EM=\frac{2\times \varepsilon \times \xi \times f\left(ka\right)}{3\eta }$$
(3)$$\xi =\frac{\eta *EM}{\varepsilon }$$ where
ξ is the zeta potential, EM is the electrophoretic mobility, ε is the water dielectric constant, f(ka) is Henry’s function, and η is the medium’s viscosity.

The vesicle’s mean hydrodynamic diameter (D_h_), size distribution (PdI), and
ξ (mV) were measured in 1 mL buffer in the presence of increasing peptide concentration. The lipid concentration was fixed at 100 μM, while the peptide concentration usually varied from 0 to 24 μM.

#### 2.2.7. 5(6)-Carboxyfluoresceine (CF) Incorporation into LUVs

For the CF leakage assay, 25 mM (total lipid) LUVs in 150 μL were prepared in 10 mM Tris-HCl buffer, pH 7.4, containing 50 mM CF. Non-encapsulated CF (~0.13 mL sample) was separated by size exclusion chromatography using a pre-packed Sephadex G-25 filter column (GE Healthcare, Buckinghamshire, UK) of 5 × 1.5 cm, equilibrated with 10 mM Tris-HCl buffer, pH 7.4, with 300 mM NaCl. Columns were previously saturated with LUVs prepared with lipids of the same composition as those containing CF. The collected LUV suspension containing CF (~1.5 mL) was quantified by phosphate determination [38].

#### 2.2.8. LUV Permeabilization by Peptides: Leakage of 5(6)-Carboxyfluorescein (CF)

CF leakage assays were performed in a 96-well plate with a black bottom, in which a serial peptide dilution was performed. The peptide solution was prepared at twice the final desired concentration in 75 μL of 10 mM Tris-HCl buffer, pH 7.4, containing 300 mM NaCl. These solutions were mixed with the same volume (75 μL) of a CF-containing 40 μM LUV suspension. The final peptide concentration varied from 32 μM to 0.125 μM, and the final lipid concentration was 20 μM in all wells. Peptide-induced vesicle permeabilization triggers CF leakage. When excited at 490 nm, the CF maximum emission fluorescence at 520 nm increases upon dilution. Emission was recorded for 60 min at 37 °C in a BioTek Synergy HT microplate reader (Bio-Tek, Winooski, VT, USA). The emission intensity of total LUV permeabilization was measured after adding 1.5 μL of a 10% (*v*/*v*) polidocanol solution to each well (positive control), while the buffer solution served as the negative control. The percentage of CF leakage was calculated using Equation (4):

CF leakage (%) = 100 × (F_P_ − F_0_)/(F_T_ − F_0_)
(4)

where F_P_ is the fluorescence intensity after 60 min, and F_0_ and F_T_ are the fluorescence intensities of the negative (before peptide addition) and positive controls (after addition of polidocanol, 100% permeabilization), respectively.

The leakage percentage was analyzed as a function of peptide concentration [P] by adjusting the Hill equation to the experimental data. This allowed us to calculate the peptide concentration necessary to lyse 50% of the LUV (C_50_) and estimate the process cooperativity (n).

% Leakage = 100 × [P]^n^/(C_50_^n^ + [P]^n^)
(5)


#### 2.2.9. Optical Microscopy of Giant Unilamellar Vesicles (GUVs)

GP- and RBC-GUVs were grown with the electroformation method [39]. Briefly, 8 μL of lipid chloroform solution (2 mM total lipid) was spread on the surfaces of two glass slides coated with fluor-tin oxide layers, and a stream of N_2_ was used to ensure that the organic solvent was evaporated. The two glasses, separated by a Teflon spacer, were assembled to form a chamber filled with 0.2 M sucrose and connected to a function generator with an AC field of 1 V and a frequency of 10 Hz for 30 min. GN-GUVs were grown with the polymer-assisted method [40]. A thin layer of polyvinyl alcohol (PVA) solution was spread on two glass slides and left in an oven at 60 °C to dry. Then, 8 μL of a lipid chloroform solution (2 mM total lipid) was spread on the polymer cushion and dried under a stream of N_2_. The two glasses, separated by a Teflon spacer, were assembled to form a chamber filled with 0.2 M sucrose. The GUVs were collected after 2 h. A homemade observation chamber was filled with 95 μL of the desired peptide concentration prepared in 0.2 M glucose. Then, 5 μL of the GUV dispersion in 0.2 M sucrose was added, the chamber was closed with a cover slip, and observations started immediately. Phase contrast image sequences were acquired with a PCO.edge 4.2 sCMOS digital camera (EXCELITAS, Kelheim, Germany) attached to an inverted Zeiss Axiovert 200 microscope (Baden-Württemberg, Germany) equipped with 40× and 63× objectives.

#### 2.2.10. Minimum Inhibitory Concentration (MIC) Determination

The peptides’ MICs against several bacterial species were determined following a known protocol [41]. Each assay was conducted in triplicate. Initially, 5 mL of Mueller–Hinton broth (MHB) was inoculated with a small number of bacteria and incubated overnight at 37 °C. The cell suspension was then diluted 50 times in an MHB medium and incubated under the same conditions until the optical density (O.D.) at 600 nm reached 0.4. The suspension was further diluted 250 times in an MHB medium, resulting in a bacterial suspension of approximately 10^6^ CFU/mL. In a 96-well polypropylene plate, a serial dilution of the peptides was performed at twice the desired final concentration in an MHB medium, ranging from 32 μM to 0.0625 μM, to a final volume of 50 μL. Then, 50 μL of bacterial suspension was added to each well, resulting in a final concentration of approximately 5 × 10^5^ CFU/mL. To verify the final bacterial concentration in the 96-well plate, a 1000-fold dilution was made from approximately 10^6^ to 10^3^ CFU/mL, and 10 μL of this dilution was plated on LB-agar plates. The plates were incubated at 37 °C for 18 h, after which the number of colonies was counted and normalized to determine the actual concentration in the assay.

An MHB medium inoculated with 50 μL of the bacterial suspension was used as a positive control, and a sterile MHB medium was used as a negative control. After 18 h, bacterial growth was visually assessed to determine the MIC.

#### 2.2.11. Hemolytic Activity (HA)

The HA of BP100, R^2^R^5^-BP100, and R^2^R^5^-BP100-A-NH-C_16_ was assessed using the method outlined by Oddo and Hansen [42]. Five milliliters of human blood, sourced from healthy volunteers and collected in heparinized tubes to prevent coagulation, were mixed with 40 mL of sterile phosphate-buffered saline (PBS) (140 mM NaCl, 10 mM phosphate, pH 7.4). The mixture was then centrifuged at 1500 rpm (500× *g*) for 10 min. After centrifugation, the supernatant was discarded, and the washing process was repeated three times. Then, 9.7 mL of PBS was added, creating a 3% RBC suspension. Serial dilutions of the peptides in PBS were prepared in a 96-well polypropylene plate (Corning, NY, USA), with final concentrations ranging from 128 to 0.25 μM and a total volume of 50 μL per well.

For the hemolysis assay, 50 μL of 3% RBC suspension was added to each well, resulting in a final RBC concentration of 1.5%. Sterile PBS served as the negative control, while 0.1% (*v*/*v*) Triton X-100 was used as the positive control to induce complete hemolysis. The plate was incubated with shaking at 37 °C for 3 h, followed by centrifugation at 1200 rpm (400× *g*) for 10 min. Supernatants (50 μL) containing released hemoglobin were transferred from each well to a flat-bottomed 96-well polystyrene plate (Greiner Bio-One, Kremsmünster, Austria). Absorbance was measured at 540 nm using a Synergy HT plate reader (Bio-Tek, Winooski, VT, USA). Hemolysis (%) was calculated using the absorbance data with the following formula:

Hemolysis (%) = (100 × (A_PEP_ − A_NEG_))/(A_POS_ − A_NEG_)
(6)

where A_PEP_, A_NEG_, and A_POS_ refer to the sample’s absorbance with the peptide (A_PEP_) and the negative, A_NEG_, and positive, A_POS_, controls, respectively.

## 3. Results

We studied the interaction of BP100 and its analogs, R^2^R^5^-BP100 and R^2^R^5^-BP100-A-NH-C_16_, with model membranes mimicking the lipid composition of GP and GN bacteria and of human RBCs (Materials and Methods, Table 2), as well as their antimicrobial and hemolytic properties, to understand the mechanisms that modulate the interaction of the peptide with target membranes.

### 3.1. Monitoring Peptide Binding via Tyr Fluorescence

Tyrosine fluorescence emission is affected by medium polarity, increasing upon membrane binding [43,44]. The emission intensity of the peptide at 305 nm was measured as a function of membrane concentration to assess the peptide’s affinity for the different model membranes. The percentage of bound peptide was calculated according to Equation (1) (Section 2.2.4).

Fluorescence intensity of BP100 increased with increasing GP (Figure 1) and GN (Figure 2) LUV concentration, saturating at a [Lipid]/[Peptide] (L/P) ratio of approximately 5 in GP (Figure 1B) and 20 in GN-LUVs (Figure 2B). A similar behavior was observed for R^2^R^5^-BP100, with saturation occurring at [L]/[P] ratios of about 5 for GP (Figure 1D) and 8 for GN (Figure 2D). With RBC-LUVs, the fluorescence of BP100 and R^2^R^5^-BP100 decreased due to extensive light scattering (Figure 3B,D, respectively). In contrast, the fluorescence intensity of R^2^R^5^-BP100-A-NH-C_16_ increased with GP, GN, and RBC-LUVs (Figure 1F, Figure 2F, and Figure 3F, respectively).

Peptide binding reached a plateau in all systems, although at variable [Lipid]/[Peptide] (L/P) molar ratios (Figure 1, Figure 2 and Figure 3). For GP and GN, saturation was reached in the region where the charge of the lipid–peptide system approached electroneutrality. Regarding RBCs, although their phospholipid components carry charged polar head groups, these lipids, and, therefore, membranes, are electrically neutral.

To assess the peptides’ affinity for the model membranes, the emission intensity at 305 nm was plotted as a function of the [Lipid]/[Peptide] molar ratio. Table 3 presents a quantitative analysis of the peptide’s affinity towards the membrane models, calculated as the inverse of the L/P ratio at which the peptide was 50% bound (P_50_/L) (Figure 1, Figure 2 and Figure 3). The higher this ratio, the lower the affinity. While BP100 and R^2^R^5^-BP100-A-NH-C_16_ presented a greater affinity for GP than for GN, the affinity of R^2^R^5^-BP100 for both membranes was similar. When comparing peptides, the affinity varied; in the case of GP, the order was BP100 ≈ R^2^R^5^-BP100-A-NH-C_16_ > R^2^R^5^-BP100, while in the case of GN, it varied in the order BP100 < R^2^R^5^-BP100 ≈ R^2^R^5^BP100-A-NH-C_16_. The order of the values indicates that the replacement of Lys by Arg residues favored the interactions of the peptides with the GN-LUVs by establishing guanidine–phosphate hydrogen bonds.

### 3.2. Effect of LUVs on Peptide Conformations

CD spectra of peptides in solution in the absence or presence of GP, GN, and RBC-LUVs are presented in Figure 4.

Typically, an α-helical conformation yields CD spectra with a maximum at 190–195 nm and a double negative peak with similar intensities, with minima at 208 nm and 220 nm [45]. Solution spectra of BP100 (Figure 4A) and R^2^R^5^-BP100 (Figure 4B) indicate that, in the absence of LUVs, these peptides were disorganized. In contrast, the R^2^R^5^-BP100-A-NH-C_16_ spectrum suggests some degree of peptide organization, possibly due to aggregation triggered by the long acyl chain, leading to a micellar-type arrangement (Figure 4C). The spectrum of RBC-bound R^2^R^5^-BP100-A-NH-C_16_ (Figure 4C, blue line) is the only one that fulfills the description of a typical α-helix. In previous work, we demonstrated that BP100 acquires an α-helical conformation upon binding to POPC: POPG LUVs [24], in agreement with other studies [46]. For BP100, the presence of RBC-LUVs did not affect the peptide conformation (Figure 4A).

Several phenomena can lead to distortions of CD spectra; among those, light scattering due to aggregation can cause shifts in the maxima and minima to longer wavelengths and spectral flattening, as well as loss of the negative peak intensity at ca. 208 nm when compared to the peak intensity at 222 nm [45,47]. Except for all three peptides in the presence of RBCs, all spectra of membrane-bound peptides displayed these features to a greater or lesser extent (Figure 4), suggesting peptide-promoted vesicle aggregation. This is better observed in the CD spectrum of 200 μM R^2^R^5^-BP100 in the presence of 1.00 mM RBC-LUVs (Appendix A).

BP100 yielded similar spectra in both GP and GN (Figure 4A); the same was seen for R^2^R^5^-BP100 (Figure 4B). The spectra of both peptides were different (compare Figure 4A and Figure 4B). The maxima and minima in Figure 4B were further shifted to longer wavelengths, with those in GP’s spectrum being more shifted than in GN’s. For R^2^R^5^-BP100, the minimum in the 210 nm region (Figure 4B) decreased more than that of BP100 (Figure 4A), and the spectra were flattened when compared to those in Figure 4A (see ellipticity values in the ordinates). These effects were even more pronounced for R^2^R^5^-BP100-A-NH-C_16_ in the presence of both GP and GN (Figure 4C). The results suggest that Lys/Arg replacement improves peptide binding to membranes because the Arg’s guanidinium group binds strongly to the lipid phosphate moiety.

### 3.3. Effect of Peptide Binding on Vesicle Diameter (Dh), Polydispersity (PDI), and Zeta Potential (ζ): Dynamic Light Scattering (DLS) Studies

The effect of peptides on the apparent hydrodynamic diameter (Dh), polydispersity index (PdI), and zeta Potential (ζ) was examined using dynamic light scattering (DLS) (Figure 5, Figure 6 and Figure 7). The setup of the DLS experiments was different from those measuring fluorescence, where increasing LUV concentrations were added to a fixed peptide concentration. In contrast, in the DLS experiments, increasing peptide concentrations were added to a fixed membrane concentration.

As GPs are highly charged, one would expect that the relative peptide/lipid ratio required for charge neutralization and subsequent aggregation would be high. For BP100 (Figure 5A) and R^2^R^5^-BP100 (Figure 5B), the increase in Dh was small up to 15 μM peptide, indicating non-significant aggregation. The values of PdI remained low. Similar behavior was observed for R^2^R^5^-BP100-A-NH-C16, except that these changes began at a peptide concentration of 20 μM.

The variation in zeta potential (ζ) with peptide concentration was similar for the three peptides, remaining constant at around −30 and −40 mV up to 15 μM with BP100 and R^2^R^5^-BP100 (Figure 5A,B) and 20 μM for R^2^R^5^-BP100-A-NH-C_16_ (Figure 5C). The value of ζ remained negative up to 24 μM peptide (Figure 5). With R^2^R^5^-BP100-A-NH-C_16_, D_h_ increased with the addition of peptide (Figure 5C), indicating that a higher binding of R^2^R^5^-BP100-A-NH-C_16_ to GP-LUVs leads to a higher decrease in the negative charge of the LUVs and significant aggregation.

GN contains a smaller negative surface charge than that of GP-LUVs. DLS indicated vesicle aggregation in a peptide concentration-dependent mode for GN-LUVs (Figure 6).

The effects of the peptides on Dh, PDI, and ζ were significantly different in GN-LUVs, displaying an initial average size of 138 nm and zeta potential of −35 mV. BP100 and R^2^R^5^-BP100-A-NH-C_16_ promoted an increase in the LUV ζ at concentrations above 6 μM, where the zeta potential became positive (Figure 6). For R^2^R^5^-BP100, the zeta potential remained negative up to 16 μM (Figure 6). Increasing concentrations of the three peptides led to an increase in Dh and PdI, when the zeta potential reached a value of zero, becoming positive thereafter. The electrostatic interaction between the peptides and LUVs resulted in surface charge neutralization, a decrease in vesicle repulsion, and subsequent aggregation, as indicated by the increased solution turbidity.

RBC-LUVs had an average size of 140 nm with a polydispersity index (PdI) of 0.1 and a negative ζ-potential of approximately −15 mV, reflecting the lipid composition of this membrane model and possibly the binding of buffer ions to phospholipids. With 1 μM R^2^R^5^-BP100-A-NH-C_16_, RBCs already showed a positive potential with an increase in Dh, which stabilized at ca. 200 nm, with PdI close to 0.3 (Figure 7C). The zeta potential results indicated that R^2^R^5^-BP100-A-NH-C_16_ bonded efficiently to all LUVs.

The zeta potential of RBC-LUVs starts at values around −15 mV and reaches neutrality with the formation of micrometric vesicles, exhibiting high polydispersity. The DLS data for vesicles with R^2^R^5^-BP100-A-NH-C_16_ indicate effective insertion into the RBC-LUVs, with a potential difference of +40 mV and diameters of less than 200 nm.

### 3.4. Peptide-Promoted LUV Permeabilization

To estimate the rate and extent of peptide-induced leakage from the internal vesicle compartment contents, we measured the fluorescence increase in LUV-encapsulated CF (Figure 8). The fluorescence of vesicle-trapped CF is self-quenched at the concentration used here (see Methods 2.2.8). Upon permeabilization and CF dilution in the external aqueous medium, fluorescence increases, providing a clear and precise indication of membrane permeabilization. All three peptides induced GP-LUV permeabilization (Figure 8A). With GP-LUVs, the peptide concentrations required to cause 50% leakage (C_50_) were similar for the three peptides (2.8 μM, 2.5 μM, and 2.4 μM for BP100, R^2^R^5^-BP100, and R^2^R^5^-BP100-A-NH-C_16_, respectively) (Table 4).

All three peptides produced 100% leakage of GP-LUVs (Figure 8A), while for GN-LUVs, maximum permeabilization was around 70% (Figure 8B). In addition, while C_50_ values were similar for the three peptides (ca. 2.6 μM) in GP, for GN-LUVs, these values were different (Table 4).

Note that the NaCl concentration in the LUVs’ external solution, necessary to equilibrate the addition of 50 mM CF in the internal compartment of the vesicles, was 300 mM (Figure 8). This salt concentration essentially eliminates the importance of electrostatic interactions but enhances the hydrophobic effect. These considerations rationalize the near identity of the three peptide-promoted GP-LUV leakage with the most charged LUVs, illustrate the importance of the Lys/Arg replacement in the less charged GN-LUVs, and point to the relevance of the hydrophobic effect with RBC-LUVs.

RBC-LUV permeabilization can be related to cytotoxicity in eukaryotic cells, which is not a desired property for antimicrobial agents. BP100 and R^2^R^5^-BP100 showed low efficiency in permeabilizing these membranes (Figure 8C), with approximately 30% permeabilization at 32 μM BP100 and 70% permeabilization at 20 μM R^2^R^5^-BP100. In contrast, R^2^R^5^-BP100-A-NH-C_16_ was highly efficient, causing approximately 95% leakage at concentrations above 2 μM (Figure 8C). These data align with the low extent of BP100 and R^2^R^5^-BP100 binding to RBC model membranes, as suggested by CD (Figure 4) and fluorescence (Figure 3) spectra.

R^2^R^5^-BP100-A-NH-C_16_ was also more efficient than the two other peptides in permeabilizing GN-LUVs, demonstrating that the additional hydrocarbon chain played a role in anchoring the peptide to the membrane and, very likely, modifying its mechanism of action. The same phenomenon occurred with RBC-LUVs, where C_50_ for R^2^R^5^–BP100-A-NH-C_16_ was lower than that for the other peptides. In the case of RBC-LUVs, the hydrocarbon chain is crucial for membrane binding, as these membranes do not bear a charged surface, which prevents electrostatic contributions to binding.

### 3.5. Peptide Effects on Giant Unilamellar Vesicles

Optical microscopy of giant unilamellar vesicles (GUVs) allows direct observations of the effects caused by AMPs on membranes [48,49], especially in differentiating membrane permeabilization [50] from disruption/burst [51]. Here, we use this approach to qualitatively assess the mechanism of action of BP100 and its analogs on GUVs with the same composition as GP-, GN-, and RBC-LUVs, as previously done for BP100 and other more hydrophobic analogs interacting with membranes of different lipid composition [24,52]. Figure 9 shows representative image sequences of the GUV compositions used in this work in the presence of BP100 and its two analogs. The dominant effect caused by the three peptides on GP- and GN-GUVs was vesicle collapse or burst at relatively low peptide concentrations (2–3 μM, similar to the MIC values listed in Table 5), as exemplified in Figure 9A,B. Vesicle bursting was also the mechanism observed previously for BP100 against anionic PC/PG membranes [24]. Additionally, the peptides often induced the adhesion of GUVs to the glass substrate and formation of dense spots on the GUVs’ surface, as shown in Figure 9B, for instance, in the presence of BP100 and R^2^R^5^-BP100. On the other hand, the effects of BP100 and its analogs on RBC-GUVs were very mild, as most GUVs remained intact even after the addition of 50–100 μM peptide. However, vesicle adhesion onto the glass (last images in Figure 9C in the presence of the analogs), vesicle aggregation (third snapshot in Figure 9C in the presence of BP100 (top row) and second snapshot in Figure 9C in the presence of R^2^R^5^-BP100 (middle row)), and vesicle collapse (first sequence in Figure 9C in the presence of R^2^R^5^-BP100-A-NH-C_16_ (bottom row)) were occasionally observed. In the case of RBC-LUVs, the C_16_ moiety is crucial for membrane binding since these membranes do not bear a charged surface. Membrane permeabilization was also found for BP100-Ala-NH-C_16_H_33_, an R2R5-BP100-Ala-NH-C16H33 analog lacking the Lys/Arg replacement [37].

Taken together, the qualitative results obtained by optical microscopy indicate that the peptides interact more strongly with GN- and GP-GUVs, causing vesicle collapse, which is compatible with a carpet mode of action, similar to that previously observed for BP100 interacting with PC: PG membranes [24]. Vesicle adhesion/aggregation to the glass and formation of dense spots occurred quite often for all membrane mimetics employed, which can be explained by the peptide’s ability to bridge opposing bilayers, either from neighboring vesicles, as observed also for LUVs (Figure 5, Figure 6 and Figure 7), or folding of the membrane of a single vesicle, as discussed earlier [49].

### 3.6. Determination of Minimum Inhibitory Concentration, MIC, in Bacteria

*E. coli*, *S. aureus*, and *B. subtilis* were selected to assay the peptide’s antimicrobial properties. The minimum inhibitory concentration (MIC) was determined as described in Section 2.2.10, and the results are presented in Table 5. Comparing the MIC values to those existing in the BP100 literature [23,52], the MIC of the *S. aureus* N315 species was different. The MICs of peptides were determined for two *S. aureus* strains, N315 and ATCC 25923 (Table 5), yielding different MIC values. This suggests that other components of the strains may be responsible for the varied responses, even for peptides with very low molecular weights.

R^2^R^5^-BP100-A-NH-C_16_ was less efficient as an antibacterial agent, despite the additional hydrophobic interaction (Table 5). The preferential insertion of the alkyl chain in the bacterial lipid membrane probably modified the peptide moiety’s interaction with the membrane. Additional hydrophobicity did not increase the bactericidal action of other BP100 analogs [37]. However, different bacterial strains present differential sensitivity to the peptides, as seen in the case of varying *Staphylococcus aureus* strains (Table 5).

The results of experiments on peptide-induced membrane permeability (Table 4) led to the expectation of an enhancement in the antimicrobial activity of the hydrophobic R^2^R^5^-BP100-A-NH-C_16_ peptide compared to BP100 and R^2^R^5^-BP100. Still, the observed results were the opposite, with MIC values in the 16–32 μM range for R^2^R^5^-BP100-A-NH-C_16_ and 1–16 μM for either BP100 or R^2^R^5^-BP100 (Table 5).

### 3.7. Peptide Effect on Mammalian Cells: Red Blood Cell Hemolysis

To evaluate the peptide’s action on mammalian cells, its effect on RBC stability was assessed. The extent of hemolysis was marginally greater for R^2^R^5^-BP100 compared to BP100, and at 64 μM, both peptides produced ca. 15% of hemolysis (Figure 10). With R^2^R^5^-BP100-A-NH-C_16_, however, almost 100% of the RBCs were destroyed at 64 μM, clearly showing the effect of the alkyl chain on the RBC membrane.

## 4. Discussion

Considering the complexity of biological membranes, work with model systems contributes to the understanding, at the molecular level, of events occurring during the antimicrobial peptide–membrane interaction process. The present study demonstrates how differences in membrane lipid composition influence this interaction.

A multi-technique approach was used to examine the effects of the interaction between the AMP BP100 and two of its analogs, R^2^R^5^-BP100 and R^2^R^5^-BP100-NH-C_16_, on structural and functional properties of model membranes consisting of the main lipids of GP and GN bacteria, as well as those of mammalian RBCs. Additionally, the effect of their activity on GP, GN, and RBCs was assessed. The results provided a detailed picture of the influence of peptide nature and membrane composition on several membrane properties.

Lipid composition and membrane net charge influenced the peptide’s interaction with LUV and GUV model membranes. Fluorescence experiments demonstrated the binding of all three peptides to negatively charged GP and GN (Figure 1 and Figure 2, respectively), reaching saturation at different L/P molar ratios and revealing a significant contribution of electrostatic interactions. The fluorescence data also indicate that BP100 and R^2^R^5^-BP100 caused extensive RBC aggregation. R^2^R^5^-BP100-A-NH-C_16_ bonded to GP, GN, and RBCs.

Lys/Arg replacement in BP100, yielding R^2^R^5^-BP100, did not result in significant conformational changes, as indicated by the similarity in their circular dichroism (CD) spectra. Despite their conformational similarities, differences were observed, as described in the Results section. Differences in the peptide’s biological activity were demonstrated in the MIC assay, where R^2^R^5^-BP100 was more active than BP100 toward *S. aureus* and *B. subtilis*. In a study by Torcato et al. [23], which utilized R-BP100, an analog in which Arg replaced all Lys residues, this peptide demonstrated both enhanced biological activity and improved interaction with lipid membranes. Our results, as well as those of Torcato et al. [23], suggest that the Arg guanidine group allows more effective interactions with lipid phosphate groups.

CD spectra (Figure 5) showed that, whereas BP100 and R^2^R^5^-BP100 solution conformation is disordered, all three peptides acquired secondary structure in the presence of GP and GN, as observed previously for BP100 [24]. In the case of GP and GN, electrostatic effects also play a fundamental role in determining that, when the system’s electroneutrality is reached, vesicle aggregation occurs, as shown by CD spectra. In the DLS experiments, electroneutrality was not achieved in the study with GP. With GN, Dh increased for all three peptides in the region of electroneutrality. Higher PdI values showed that the aggregates were heterogeneous, and an increase in zeta potential also indicated binding. Ferreira et al. reported that a BP100 analog containing Trp at the N-terminus promoted vesicle aggregation [53]. Interestingly, the increase in zeta potential was much more pronounced upon the addition of R^2^R^5^-BP100-A-NH-C_16._ According to Freire et al., the increase in zeta potential is proportional to the peptide’s partition coefficient. Thus, the acyl chain in R^2^R^5^-BP100-A-NH-C_16_ acts as an anchor, strengthening peptide binding. If peptide binding occurs to the extent that the overall particle charge becomes positive, repulsion may occur, leading to particle disaggregation.

Optical microscopy of GP- and GN-GUVs revealed that the dominant effect caused by the peptides was vesicle collapse or burst, in agreement with the mechanism observed for BP100 against anionic PC/PG membranes [24], i.e., the carpet mode of action [54]. It is noteworthy that the peptides often induced GUV adhesion to the glass substrate and the formation of dense spots on the GUV surface, indicating that the peptides are capable of bridging bilayers from two different vesicles. Thus, vesicle aggregation is mediated by electroneutrality and peptide interaction with neighboring vesicles. Differences in the extent of aggregation in the peptides’ CD spectra, both qualitative and quantitative, indicate their varying ability to promote the bridging effect. This is corroborated by the DLS data, which highlight the substantial impact of the acyl chain addition to R^2^R^5^-BP100-A-NH-C_16_.

Our model membranes (RBCs) consisted of the major zwitterionic lipids (PC, PE, and SM) and uncharged lipids (CHOL) found in human RBCs. Fluorescence, CD, and DLS data provided evidence for BP100 and R^2^R^5^-BP100-promoted RBC aggregation, while this phenomenon was not observed for R^2^R^5^-BP100-A-NH-C_16_. In this latter case, the CD spectrum of an α-helix revealed peptide (homogeneous) dispersion in the bilayer. This was corroborated by a slight increase in Dh at the concentration of 1 μM peptide and an increase in zeta potential. We ascribe aggregation to peptide bridging of two (or more) vesicles. This is supported by the GUV results, which show two vesicles frequently found next to each other in the presence of BP100 and R^2^R^5^-BP100.

As RBCs do not carry a net charge, how can peptide–membrane interaction be rationalized? We propose that PE is responsible for this process. PE, although being zwitterionic, possesses physicochemical and structural properties that are quite different from those of PC, the main one being the former’s ability to form intermolecular hydrogen bonds. The network formed by these hydrogen bonds at the bilayer headgroup region is responsible for the approximately 20 degrees higher gel-to-liquid crystal phase transition temperature of PEs compared to PCs with the same acyl chain composition. This ability of PEs to form hydrogen bonds may contribute to the interaction between the peptides and the interface, enabling vesicle aggregation. Preferential binding of AMP to PE has been reported [55].

Regarding GP, all three peptides yielded similar C_50_ values. The GN membrane composition resulted in differentiation between the peptides, with effectiveness following the order BP100 < R2R5-BP100 < R^2^R^5^-BP100-A-NH-C_16_, indicating that Lys/Arg replacement improved peptide binding to these membranes. GUV experiments showed that vesicle bursting occurred for all three peptides. However, C_50_ values were not in agreement with DSL data; in this latter case, the peptide concentrations for the onset of changes showed a good correlation with the electroneutrality criterion for all three peptides, whereas an excess of positive charges was present in leakage experiments in the case of BP100 and R2R5-BP100, but not R^2^R^5^-BP100-A-NH-C_16_. These results strongly suggest a different mechanism of action for R^2^R^5^-BP100-A-NH-C_16_, possibly related to the fact that, while BP100 and R^2^R^5^-BP100 have a surface location, R^2^R^5^-BP100-A-NH-C_16_ is anchored in the membrane acyl chain region.

In the case of RBCs, the absence of surface charge precludes the participation of electrostatic interactions in the mechanism of R^2^R^5^-BP100-A-NH-C_16_-promoted leakage. Thus, leakage at such a small C_50_ (0.09 μM) value suggests a detergent-like action of the peptide. This concentration corresponds to 0.5% (in moles) of lipid and is in good agreement with DLS data, which showed a small peak in Dh at 1% peptide concentration. The results from GUVs corroborate the lack of vesicle aggregation. Although the data demonstrate that BP100 and R^2^R^5^BP100 are essentially ineffective in causing CF leakage, fluorescence, CD, and DLS, the GUV data show that the peptides are capable of aggregating RBC vesicles. It is very likely that the driving mechanism is an interaction between the peptides and POPE at the membrane surface that is strong enough to cause peptide-promoted inter-vesicle bridging. The occurrence of two vesicles bound together is also observed more frequently in GUV images of BP100 in the presence of RBCs. It is noteworthy that this configuration occurs without eliciting vesicle leakage.

BP100 and R2R5-BP100 were considerably active against both GP and GN bacteria. More studies are required to correlate the results obtained for model systems with those for bacteria, as the latter are much more complex entities, and their organization varies with metabolic conditions, among other factors. MIC values of BP100 and R^2^R^5^-BP100 were lower than those of R^2^R^5^-BP100-A-NH-C_16_. In contrast, the two peptides essentially lacked hemolytic activity, whereas the long-tail-containing analog was highly effective. Thus, the results obtained for R^2^R^5^-BP100-A-NH-C_16_ are undesirable from both therapeutic and toxicity points of view. Although similar results have been obtained for other AMPs, several studies have focused on lipopeptides as therapeutic agents.

## 5. Conclusions

We examined the interaction between BP100 and its analogs containing Lys/Arg replacements at residues 2 and 5 (R^2^R^5^-BP100), as well as R^2^R^5^-BP100-A-NH-C_16_, and model membranes consisting of the main lipids present in GP and GN bacteria and RBCs. Additionally, the effects of the peptides on bacteria and RBC hemolysis were examined. Peptide–membrane interactions were studied using various techniques, including fluorescence, circular dichroism (CD), dynamic light scattering (DLS), GUV optical microscopy, and dye leakage. These approaches unveiled the multiplicity of events resulting from the interactions, with both peptide structure and membrane composition influencing the results. Lys/Arg replacement did not alter the peptide conformation, albeit it did promote greater peptide binding to model membranes and increased leakage from GN. GUV data indicate that both BP100 and R^2^R^5^-BP100 caused bilayer rupture by the same mechanism. While both peptides were equally ineffective in inducing hemolysis, the latter presented a lower minimum inhibitory concentration (MIC) value against two GP bacteria tested.

On the other hand, the behavior of R^2^R^5^-BP100-A-NH-C_16_ was different from that of the two other peptides: it bonded more strongly to all model membranes due to the hydrocarbon chain intercalation in the bilayer hydrophobic portion, it caused leakage from GN and RBCs at much lower concentrations, and, in contrast to the two other peptides, it was capable of inducing hemolysis. Interestingly, all three peptides exhibited similar C_50_ values for GP leakage, indicating that, in this case, electrostatic interactions were highly predominant, and specificity did not play a role in the peptide’s mechanism of membrane bursting. The peptide-promoted GP and GN burst, aggregation, and leakage resulted from the initial surface charge neutralization of GP and GN. In contrast, BP100 and R^2^R^5^-BP100 led to RBC aggregation without vesicle lysis. Also, different from these peptides, the MIC values for R^2^R^5^-BP100-A-NH-C_16_ were higher, indicating that the change in its mechanism of action renders this peptide’s antimicrobial action less effective. This detailed assessment of the events modulating AMP-membrane interaction should contribute to the development of more effective AMPs.

## Data Availability

The original contributions presented in this study are included in the article/Appendix A. Further inquiries can be directed to the corresponding author(s).

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
