# Peer review of "Structural and Functional Effects of the Interaction Between an Antimicrobial Peptide and Its Analogs with Model Bacterial and Erythrocyte Membranes"

_biomolecules, 2025, doi:10.3390/biom15081143_

Round 1

Reviewer 1 Report

Comments and Suggestions for Authors

The authors investigated the structural and functional interactions of the antimicrobial peptide BP100 and two of its analogs, R2R5-BP100 (in which Lys residues were replaced by Arg) and R2R5-BP100-A-NH-C16 (with an Ala residue and a hydrophobic C16 tail added to the C-terminus), using model membranes that mimics Gram-positive bacteria, Gram-negative bacteria and red blood cell membranes. Through fluorescence spectroscopy, circular dichroism, DLS, microscopy and biological assays, they evaluated how these modifications influenced peptide binding, membrane disruption and antimicrobial or hemolytic activities. Overall, the study highlighted how targeted structural modifications modulate membrane selectivity and biological performance of antimicrobial peptides.

-Authors should ensure that the references are formatted according to the style guidelines of Biomolecules. Otherwise, they should be updated accordingly.

-Several abbreviations (e.g. “AMP”) have been introduced multiple times, which disrupts the overall readability. Please review and ensure abbreviations are introduced only once.

-In lines 25-26, the abbreviations should be written clearly such as Large unilamellar vesicles (LUVs) and giant unilamellar vesicles (GUVs).

-In lines 37-38, the bacteria names should be written as full name and italic form.

-In line 37, correct abbreviations should be used for Gram-positive and negative bacteria.

-In line 54, Gram-positive and negative bacteria should be written together with abbreviations.

-Please check all abbreviations in the whole article for proper reading.

-Aminoacid names should be written as full at the first usage in the article. Please check it.

-Line 78, please remove the extra spaces.

-The resolution and quality of panel C in Scheme 1 is insufficient. Please update it with a higher-quality image.

-Line 560, Please ensure consistency in font size and style across the entire manuscript.

-Line 570, In an article references should be separated by a single space on each side to maintain clarity. Please update accordingly.

-Lines 631 and 634, Latin phrases such as “et al.” should be italicized. Revise accordingly.

-In line 676, why the approximately is italic? If it is noncorrect, please correct it.

-Please include an “Author Contributions” section at the end of the manuscript.

Author Response

Reviewer 1:

The authors investigated the structural and functional interactions of the antimicrobial peptide BP100 and two of its analogs, R2R5-BP100 (in which Lys residues were replaced by Arg) and R2R5-BP100-A-NH-C16 (with an Ala residue and a hydrophobic C16 tail added to the C-terminus), using model membranes that mimics Gram-positive bacteria, Gram-negative bacteria and red blood cell membranes. Through fluorescence spectroscopy, circular dichroism, DLS, microscopy and biological assays, they evaluated how these modifications influenced peptide binding, membrane disruption and antimicrobial or hemolytic activities. Overall, the study highlighted how targeted structural modifications modulate membrane selectivity and biological performance of antimicrobial peptides.

-Authors should ensure that the references are formatted according to the style guidelines of Biomolecules. Otherwise, they should be updated accordingly.

Reference style was corrected

-Several abbreviations (e.g. “AMP”) have been introduced multiple times, which disrupts the overall readability. Please review and ensure abbreviations are introduced only once.

This was corrected throughout.

-In lines 25-26, the abbreviations should be written clearly such as Large unilamellar vesicles (LUVs) and giant unilamellar vesicles (GUVs).

Corrected

-In lines 37-38, the bacteria names should be written as full name and italic form.

Corrected

-In line 37, correct abbreviations should be used for Gram-positive and negative bacteria.

Corrected

-In line 54, Gram-positive and negative bacteria should be written together with abbreviations.

Corrected

-Please check all abbreviations in the whole article for proper reading.

All abbreviations were defined only once

-Aminoacid names should be written as full at the first usage in the article. Please check it.

Corrected

-Line 78, please remove the extra spaces.

Corrected

-The resolution and quality of panel C in Scheme 1 is insufficient. Please update it with a higher-quality image.

Corrected

-Line 560, Please ensure consistency in font size and style across the entire manuscript.

Font size and style was corrected and uniformized.

-Line 570, In an article references should be separated by a single space on each side to maintain clarity. Please update accordingly.

Corrected

-Lines 631 and 634, Latin phrases such as “et al.” should be italicized. Revise accordingly.

Corrected

-In line 676, why the approximately is italic? If it is noncorrect, please correct it.

Corrected

-Please include an “Author Contributions” section at the end of the manuscript.

We included in our submission as follows:

Conceptualization, Marcelo Porto Bemquerer and Iolanda Midea Cuccovia; Methodology, Marcelo Porto Bemquerer, Sumika Kiyota and Magali Aparecida Rodrigues; Software, Danilo Kiyoshi Matsubara; Validation, Shirley Schreier; Formal analysis, Iolanda Midea Cuccovia; Investigation, Michele Lika Furuya, Gustavo Penteado Carretero, Marcelo Porto Bemquerer, Sumika Kiyota, Magali Aparecida Rodrigues, Tarcillo José de Nardi Gaziri, Norma Lucia Buritica Zuluaga, Marcio Nardelli Wandermuren, Karin do Amaral Riske and Iolanda Midea Cuccovia; Resources, Marcelo Porto Bemquerer, Magali Aparecida Rodrigues and Iolanda Midea Cuccovia; Data curation, Tarcillo José de Nardi Gaziri, Marcio Nardelli Wandermuren, Karin do Amaral Riske, Shirley Schreier and Iolanda Midea Cuccovia; Writing – original draft, Karin do Amaral Riske, Hernan Chaimovich and Iolanda Midea Cuccovia; Writing – review & editing, Hernan Chaimovich, Shirley Schreier and Iolanda Midea Cuccovia; Supervision, Gustavo Penteado Carretero, Hernan Chaimovich and Iolanda Midea Cuccovia; Project administration, Iolanda Midea Cuccovia; Funding acquisition, Iolanda Midea Cuccovia.

Reviewer 2 Report

Comments and Suggestions for Authors
  1. The reasons or advantages for performing substitution modifications at positions 2 and 5 of BP100 should be clearly stated.
  2. While the article focuses primarily on model experiments, a small number of experiments demonstrating interaction with bacteria could be added, such as live/dead cell staining or scanning electron microscopy (SEM).
  3. The mechanism is unclear: why does R²R⁵-BP100 perform better than BP100?
  4. In R²R⁵-BP100-A-NH-C₁₆, the acyl chain acts as an anchor enhancing peptide binding, but both activity and safety are reduced. What is the reason for this?

Author Response

  1. The reasons or advantages for performing substitution modifications at positions 2 and 5 of BP100 should be clearly stated.

The hypothesis was whether substitution of arginine for lysine residues in the same face of a α-helix (I, i+3 positions) would affect the antimicrobial activity of BP100. The reason for this hypothesis is that guanidine group of arginine establishes stronger and geometrically favored electrostatic interactions with phosphate groups than the primary amino group of lysine. Part of this answer was added to the introduction in page 3, lines 80 – 97.

  1. While the article focuses primarily on model experiments, a small number of experiments demonstrating interaction with bacteria could be added, such as live/dead cell staining or scanning electron microscopy (SEM).

Although we agree with the interest of these proposed experiments, the purpose here was exclusively to assess the effect of the peptides on the minimum inhibitory concentration. We were not focused here in investigating the mechanism of the peptide actions on the bacteria.

  1. The mechanism is unclear: why does R²R⁵-BP100 perform better than BP100? What is the reason for this?

This is discussed in page 20, lines 604 – 612 and the effct is evident in page 13 lines 365 – 367 (Table 3)

  1. In R²R⁵-BP100-A-NH-C₁₆, the acyl chain acts as an anchor enhancing peptide binding, but both activity and safety are reduced.

One can only speculate of the reason for reduced activity and safety of R²R⁵-BP100-A-NH-C₁₆ compared to the other two peptides. The high activity upon zwitterionic membranes of red-blood cells than either BP100 and R²R⁵-BP100 indicates that the hydrophobic interactions predominate when compared to the electrostatic interactions by the guanidine group of arginine residues or the amino groups of the lysine residues. Nevertheless, it is unclear why the Gram-positive and Gram-negative microorganisms were less sensitive to the hydrophobicity of the R²R⁵-BP100-A-NH-C₁₆ peptide than red-blood cell. Since R2R5–BP100-A-NH-C16 is capable of promote CF leakage of GP, GN, and RBC LUVs, the results indicate that in more complex bacterial membrane this peptide anchors in the bilayer without causing lysis, but the reason for this behavior is unclear. Chen et al. (2021) showed that an amphiphilic fourteen-residue peptide had diminished antimicrobial activity and enhanced hemolysis when its N-terminus was modified by myristoyl, palmitoyl, or stearoyl residues. In contradistinction to our results, Chen et al. (2021) observed a significant increase in hemolytic activity when the lysine residues were replaced by arginine residues.

Chen, S.P.; Chen, E.H.; Yang, S.Y.; Kuo, P.S.; Jan, H.M.; Yang, T.C.; Hsieh, M.Y.; Lee, K.T.; Lin, C.H.; Chen, R.P. A systematic study of the stability, safety, and efficacy of the de novo designed antimicrobial peptide PepD2 and Its modified derivatives against Acinetobacter baumannii. Front Microbiol. 2021, 18, 678330.

Reviewer 3 Report

Comments and Suggestions for Authors

The work is of very high scientific quality and very well organized. It uses a significant variety of biophysical and functional techniques to study the structure and action of the original peptide (BP100) and two analogues of it. Personally, as a researcher specializing in lipids and AMPs, I would think that the addition of a C16 tail to an AMP transforms it into a structure more similar to a lysophospholipid, rather than the traditional behavior of an AMP, as is actually the case, losing its specificity of action against prokaryotic and eukaryotic cells. However, conducting a literature search on the topic, I found several studies where this type of strategy has been beneficial by providing antimicrobial effects while maintaining low cytotoxicity in eukaryotic cells. This leads me to accept the authors' strategic approach as valid, although it would be prudent to add this type of analysis to the justification for choosing the analogue studied. Despite this, I also found an article (https://doi.org/10.3389/fmicb.2021.678330) that analyzes the modification of a synthetic AMP with hydrocarbon tails of different chain lengths, showing a significant increase in hemolysis as the chains extend beyond C8. Perhaps it would be prudent for the authors to consider this citation in this article and, more importantly, for future optimization analyses.
Based on all of the above, I believe the authors could make a minor revision to the justification for their strategy in choosing C16. That said, I recommend to the editor that this article be published after the aforementioned minor revisions.

Author Response

The work is of very high scientific quality and very well organized. It uses a significant variety of biophysical and functional techniques to study the structure and action of the original peptide (BP100) and two analogues of it. Personally, as a researcher specializing in lipids and AMPs, I would think that the addition of a C16 tail to an AMP transforms it into a structure more similar to a lysophospholipid, rather than the traditional behavior of an AMP, as is actually the case, losing its specificity of action against prokaryotic and eukaryotic cells. However, conducting a literature search on the topic, I found several studies where this type of strategy has been beneficial by providing antimicrobial effects while maintaining low cytotoxicity in eukaryotic cells. This leads me to accept the authors' strategic approach as valid, although it would be prudent to add this type of analysis to the justification for choosing the analogue studied. Despite this, I also found an article (https://doi.org/10.3389/fmicb.2021.678330) that analyzes the modification of a synthetic AMP with hydrocarbon tails of different chain lengths, showing a significant increase in hemolysis as the chains extend beyond C8. Perhaps it would be prudent for the authors to consider this citation in this article and, more importantly, for future optimization analyses.
Based on all of the above, I believe the authors could make a minor revision to the justification for their strategy in choosing C16. That said, I recommend to the editor that this article be published after the aforementioned minor revisions.

We thank the reviewer for alerting us to the excellent work by Chen et al. We have added the reference that is cited in page 3 line 93, 94  reference [29]

Reviewer 4 Report

Comments and Suggestions for Authors

  1. Scheme 1 C is unclear.
  2. What does the independent paragraph from lines 108 to 111 mean? It should be merged with the following paragraphs.
  3. The introduction does not clearly explain where the major problem lies and the significance of this research.
  4. This study only selected one antimicrobial peptide and its analogues. Whether the research results are universal, that is, whether the results can support the title, is questionable. More analogues should be designed for research to reflect the universality of the results.
  5. If the methods used in section 2.2.4 are borrowed from others, the references should be cited. If they are self-created methods, explanations should be provided. The same issue exists in other method sections.
  6. There should be a space between "2.2.75(6)-".
  7. Where is section 3.1?
  8. What is the calculation formula for P50/L?
  9. Why are the concentrations of the peptides used in Figures 5, 6, and 7 different?
  10. The discussion section is poorly written and lacks substantial literature support.
  11. The introduction, discussion, and conclusion sections are not concise enough.
  12. The color indication in the supplementary figures is incorrect.

Author Response

  1. Scheme 1 C is unclear.

The Scheme was improved.

  1. What does the independent paragraph from lines 108 to 111 mean? It should be merged with the following paragraphs.

The reviewer is right and the paragraph was modified.

  1. The introduction does not clearly explain where the major problem lies and the significance of this research.

We appreciate this comment but text in the introduction, especially in pages 3 – 5, lines 80 – 82, 88 – 91, 115 – 117, 129 – 133, constitute descriptions of the aims and main conclusions of this work.

  1. This study only selected one antimicrobial peptide and its analogues. Whether the research results are universal, that is, whether the results can support the title, is questionable. More analogues should be designed for research to reflect the universality of the results.

We have studied other BP100 analogs [G.P.B. Carretero, G.K.V. Saraiva, A.C.G. Cauz, M.A. Rodrigues, S. Kiyota, K.A. Riske, A.A. dos Santos, M.F. Pinatto-Botelho, M.P. Bemquerer, F.J. Gueiros-Filho, H. Chaimovich, S. Schreier, I.M. Cuccovia, Synthesis, biophysical and functional studies of two BP100 analogues modified by a hydrophobic chain and a cyclic peptide, Biochimica et Biophysica Acta (BBA) - Biomembranes, 1860 (2018) 1502–1516] and are currently obtaining other derivatives. The reviewer´s comment is timely, but in this paper we decided to include only the present data.

  1. If the methods used in section 2.2.4 are borrowed from others, the references should be cited. If they are self-created methods, explanations should be provided. The same issue exists in other method sections.

We cite previous work, but describing part of the methodology makes our work more reproducible.

  1. There should be a space between "2.2.75(6)-".

This was corrected the text now shows

2.2.7  5(6)-Carboxyfluoresceine (CF) incorporation into LUVs

  1. Where is section 3.1?

This error was corrected

  1. What is the calculation formula for P50/L?

Please see  the description in page 13 of the corrected MS, lines 363 – 372.

  1. Why are the concentrations of the peptides used in Figures 5, 6, and 7 different?

In Figures 5,6 and 7 the concentration of Lipid is fixed and the concentration of peptide varies.

  1. The discussion section is poorly written and lacks substantial literature support.

Most of the relevant literature is cited throughout the text.

  1. The introduction, discussion, and conclusion sections are not concise enough.

We would like to maintain our style and include the relevant literature.

  1. The color indication in the supplementary figures is incorrect.

This error was corrected

Round 2

Reviewer 1 Report

Comments and Suggestions for Authors

During the revision process, the authors have made the necessary changes.

Reviewer 4 Report

Comments and Suggestions for Authors

Scheme 1 C is unclear.

The author section of references 40,46 and others is different from other places.